# Effect of Some Plant-Based Substances on Microbial Content and Sperm Quality Parameters of Bull Semen

**DOI:** 10.3390/ijms24043435

**Published:** 2023-02-08

**Authors:** Aleksandar Cojkic, Ingrid Hansson, Anders Johannisson, Jane M. Morrell

**Affiliations:** 1Department of Clinical Sciences, Swedish University of Agricultural Sciences (SLU), 75007 Uppsala, Sweden; 2Biomedical Science and Veterinary Public Health, Swedish University of Agricultural Sciences (SLU), 75007 Uppsala, Sweden

**Keywords:** bull semen, pomegranate, ginger, curcumin, antibacterial effect

## Abstract

The rapid emergence of antibacterial resistance requires alternatives to antibiotics to be found, including for semen preservation. One of the possible alternatives would be to use plant-based substances with known antimicrobial effects. The objective of this study was to test the antimicrobial effect of pomegranate powder, ginger, and curcumin extract in two concentrations on bull semen microbiota after exposure for <2 h and 24 h. An additional aim was to evaluate the effect of these substances on sperm quality parameters. The bacterial count in semen was low from the beginning; however, a reduction was present for all tested substances compared with control. A reduction in bacterial count in control samples was also observed with time. Curcumin at a concentration of 5%, reduced bacterial count by 32% and was the only substance that had a slight positive effect on sperm kinematics. The other substances were associated with a decline in sperm kinematics and viability. Neither concentration of curcumin had a deleterious effect on sperm viability parameters measured by flow cytometry. The results of this study indicate that curcumin extract at a concentration of 5% can reduce the bacterial count and does not have a negative influence on bull sperm quality.

## 1. Introduction

During human history, plant extracts have been used in medicine, pharmaceutical industries and nutrition, as well as for cosmetic and other purposes. The main pharmacological activity of the plants comes from their secondary metabolites; terpenoids, such as phenolic compounds, alkaloids, and sulphur-containing compounds [1]. These chemical compounds were found to protect the plant itself against potential pathogens. Apart from this antimicrobial activity, phytochemicals possess numerous characteristics such as antioxidants, antiviral, antiparasitic, antitumor, antimutagenic, anti-inflammatory, and analgesic properties. Consumption of these plants, or their extracts, as a source of antioxidants showed promising results in patients with fertility disorders [2] as well as for the improvement of reproductive performance in animals and humans.

Good sperm quality depends on many factors that can be classified as environmental and/or endogenous in origin. One of these factors, originating from both sources, is bacterial contamination of semen. Bacterial presence in semen samples negatively influences sperm quality on many levels, decreasing sperm motility and viability, inducing premature acrosome reaction, or causing sperm aggregation. However, these effects are associated with the presence of particular microbes in semen samples [3]. The presence of these microbes in semen is unavoidable because semen collection is not a sterile process, even though strict hygiene measures are taken during semen collection. Therefore, European legislation (European Council Directive EU-2019/6) [4] requires that antibiotics must be added to commercial semen doses for artificial insemination to inhibit microbial growth. Despite achieving a reduction in bacterial growth, many antibiotics have a negative effect on sperm survival. Therefore, a combination of several antimicrobial agents at low concentrations is recommended. However, this non-therapeutic use of antibiotics in smaller doses can still lead to the development of antimicrobial resistance [5]. 

A possible alternative to adding antibiotics to the semen could be adding plant-based substances with a known antibacterial effect on different bacterial genera. Various plant extracts showed some antimicrobial properties by increasing the activity of several antioxidative enzymes. One example is pomegranate (*Punica granatum* L.) juice that at various concentrations showed a positive effect on sperm quality parameters after chilling and cryopreservation of bull semen samples [6]. Furthermore, antimicrobial activity of arils from different pomegranate varieties was observed against several bacteria [7]. Curcumin (*Curcuma longa*) extract has been shown to have a positive effect on boar spermatozoa [8], as well as on thawed goat sperm parameters [9]; it also exhibited protective properties on bull spermatozoa by the preservation of motility, mitochondrial activity, and antioxidant characteristics during induced oxidative stress [10] and during cryopreservation [11]. The antibacterial effect of curcumin on *Staphylococcus aureus* has been discussed in detail [12]. In the study by Merati and Farshad [13], adding extract of ginger (*Zingiber officinale*) resulted in improved quality and fertility of frozen-thawed ram epididymal spermatozoa. 

Schulze et al. [14] defined criteria that a potential sperm additive needs to meet to be considered as an antimicrobial alternative. These include having broad spectrum antimicrobial action, absence of sperm toxicity, no interference with fertility, high stability, high activity at common semen storage temperatures, low potential to evoke bacterial resistance, ease of application, and economic feasibility. With these criteria in mind, the objective of this study was to test the antibacterial effects of three plant-based substances; pomegranate powder, curcumin, and ginger extract, when added to bull semen extenders at different concentrations. A further aim was to evaluate the effect of the tested substances on sperm quality parameters. 

## 2. Results

Pomegranate powder (P) 5% did not affect CASA (Computer-assisted sperm analysis) parameters at Time 1, whereas ginger (G) 10% and DMSO (Dimethylsulfoxide) (D) 10% and 5% had a significant negative effect on CASA parameters (Table 1). Curcumin (C) in both concentrations, as well as P 10%, had a negative effect on some kinematic parameters although the proportion of motile sperm did not change.

After 24 h of incubation, both P samples (10% and 5%) and G 10% showed decreased values for CASA parameters compared with the control. The remaining substances did not negatively influence the CASA parameters; in fact, C 5% increased the values of most CASA parameters after 24 h of exposure (Table 2). 

Substances G 10%, G 5%, D 10%, and D 5% had a negative influence on all FC (Flow cytometry) parameters except chromatin integrity at 0 h (Table 3), apart from P 10%, which had a negative influence on mitochondrial membrane potential and chromatin integrity, and P 5%, which had a negative influence only on chromatin integrity. Curcumin 10% and 5% did not have a deleterious effect on sperm quality parameters. Similar results were observed after 24 h of exposure (Table 4). 

There was considerable variation in the number of viable aerobic bacteria (colony forming units per mL; CFU/mL) between individual bulls, although the majority (80%) of bulls had an average bacterial count lower than 100 CFU/mL before exposure to the different plant substances. Reduction in CFU with exposure time was common for all tested animals (Figure 1 and Figure 2 for Time 1 and Time 2, respectively).

A reduction in the number of CFU during exposure was also observed for all tested substances as well as controls (Table 5). 

The greatest bacterial reduction was observed after adding DMSO 5% (54%) and G 10% (42%) at Time 1. On the other hand, both P concentrations and C 10% showed the least bacterial reduction. A similar trend was seen at Time 2.

## 3. Discussion

The present study was conducted to determine whether three plant-based substances with reported antimicrobial activity could be used in semen extenders for bull semen to inhibit bacterial growth. 

Our results showed that some of the plant-based additives, pomegranate, curcumin, and ginger, had a detrimental effect on sperm motility, particularly after exposure for 24 h. Oxidative stress, which often arises during semen storage, significantly reduces sperm function and compromises sperm fertilizing ability by inducing oxidative damage to proteins, lipids, and nucleic acids [15]. A positive relationship was reported between the antioxidant capacity of seminal plasma and both sperm concentration and total motility [16]. Moreover, cryopreservation was shown to damage sperm structure and thus sperm function. Sperm storage, on the other hand, leads to the overproduction of ROS (Reactive oxygen species), thus to oxidative stress, compromising sperm integrity and fertilizing ability. Addition of a semen extender during preservation leads to a dilution of the antioxidant capability of the seminal plasma. Several compounds have been added to semen during preparation to increase antioxidative properties. Plant extracts showed promising results in the preservation of semen as a source of antioxidants; this effect is both economic and organic. Another factor that causes a decline in sperm parameters during semen storage is bacterial contamination of the semen prior to cryopreservation [17].

Pomegranate is a polyphenol-rich fruit and, as such, has been used for its antioxidative and antibacterial properties [18]. Besides its positive effect on sperm quality when added to semen extender [6] or as food supplement [19], pomegranate has been tested for its antibacterial properties against several bacteria; there was a strong correlation between total phenolic content and antimicrobial activity [20]. The level of polyphenolic compounds in pomegranates depends on the part of the plant used and which extraction method was performed. Duman et al. [7] reported that antimicrobial activity even differs between different pomegranate varieties. For these reasons, we chose to test a commercially available pomegranate powder with known active substances. In the study by El-Sheshtawy et al. [6], five concentrations (10%, 20%, 30%, 40%, and 50%) of pomegranate extract were added to a semen extender and the effect on bull semen quality parameters was evaluated. Concentrations of 10% and 20% improved frozen-thawed semen quality; the highest concentration had a deleterious effect on sperm quality. However, neither P 5% nor P 10 % had an effect on CASA parameters and there was no indication of an effect on sperm quality from the flow cytometry results. In contrast, in our study, a deleterious effect of both P 5% and P 10% on CASA parameters was noted after 24 h of exposure. Interestingly, flow cytometry results for P 5% at Time 2 (after 24 h exposure to the test substances) indicated that sperm quality parameters were preserved although they were decreased for P 10% compared with controls. 

Ginger, as an additive to semen extenders, has been reported to have a positive effect on spermatozoa, improving quality and fertility potential of ram thawed semen [13]. It had a strong antimicrobial activity against opportunistic pathogenic and multi-resistant bacteria [21,22]. However, our results showed a negative effect of ginger on both motility and viability parameters of the spermatozoa. This could be due to dissolving the ginger extract in DMSO, which also had a negative effect on sperm quality parameters in the present study. It was previously considered that DMSO would have a protective effect during sperm cryopreservation, by penetrating and dehydrating the spermatozoa to minimize intracellular ice formation [23], but it caused damage during processing of buffalo spermatozoa [24]. Increasing the DMSO concentration in the semen extender significantly decreased the proportion of intact acrosomes. The results of our study are in agreement with these findings, since G 10% and 5%, as well as D 10% and 5%, had a negative effect on all FC parameters except on chromatin at 0 h, compared with P 10% and 5%, which only negatively influenced chromatin integrity. In the study by Farshad et al. [25], DMSO concentrations were lower than in the present study, since we added enough DMSO to dissolve the ginger extract. The negative effect of ginger was probably due to the DMSO in which it was dissolved. Similar results were observed when the antibacterial effect of these substances was tested. Both G and DMSO had a strong antibacterial effect compared with other substances. 

Curcumin was previously tested as a supplement when added to semen prior to cryopreservation, based on its potent protective and antioxidative properties [8]. The same conclusion was reached in previous studies conducted on sperm samples from different animal species; curcumin had significant dose-dependent protective and antioxidative characteristics [8]. However, there was no such effect of curcumin on bull spermatozoa in our study. This could be due to the use of 70% ethanol to dissolve curcumin powder compared with other studies where DMSO was used. In the study by Tvrdá et al. [26], curcumin in different concentrations was able to prevent the decrease of some sperm quality parameters during incubation, compared with controls. However, curcumin at either concentration in our study was the only substance that did not have a negative effect on any sperm quality parameters; most of the CASA parameters were not adversely affected by C 5% after 24 h of exposure. A bacterial reduction of more than 30% was observed when C 5% at < 2 h of incubation was used. To our knowledge, there are no published results about such an antibacterial effect of curcumin in sperm samples.

The reason for the lack of a positive effect of the tested substances on sperm quality parameters could be due to our samples being tested for the first time at 24 h after semen collection. During transport, the semen samples were extended in the ratio 1.1; which could be insufficient to meet the metabolic needs of the spermatozoa, even at the low temperature. On the other hand, the condition of the packets containing the semen samples was not strictly controlled during transport to the laboratory. There is evidence that different handling conditions of human semen samples negatively affected all sperm quality parameters with time [27]. Irrespective of the extender and the storage conditions used, semen handling and preservation negatively affected sperm quality [17]. The same authors concluded that the concentration of the plant substance to be used is influenced by the extraction method and that the same extract concentration can have different effects on sperm quality due to the sperm preservation method used (refrigeration vs. cryopreservation). However, in the present study, a reduction in bacterial count (CFU/mL) with time was observed, irrespective of treatment, even though the number of bacteria was low from the beginning in most of the bull semen samples. All tested substances were reported to have an antibacterial effect on the particular cultured bacterial strains [28]; however, no information about the antibacterial effect on the same bacteria in semen samples was available. For some other plant-based substances, which were tested for their antibacterial effect in semen samples, an interspecies variation was found. An example is rosmarinic acid that was reported to have an antibacterial effect in boar semen [29,30] but did not have such an effect in bull semen [31]. Therefore, there may be some components of bull semen that prevent the antibacterial action of rosmarinic acid that is reported to occur when added to boar semen.

Bacteria negatively affect sperm quality, by competing directly with spermatozoa for nutrients supplied by the semen extender, or by the production of toxic metabolic byproducts and endotoxins [32]. Government and regional directives stipulate which antibiotics, and what concentrations, should be added to semen doses for international trade [4]. Usually, the concentrations used are lower than therapeutic ones. The main reason for this low dose and the use of several antibiotics is to avoid the toxic effect of some antibiotics on spermatozoa and to provide a wider range of antimicrobial protection. When a combination of antibiotics is used, an adequate antimicrobial effect can be achieved with no apparent toxic effect on spermatozoa [33]. However, even this sub-therapeutic dose of antibiotics can lead to the development of antimicrobial resistance. It would be interesting to see whether a combination of the plant-based substances mentioned here could achieve an antimicrobial effect without sperm toxicity. 

## 4. Materials and Methods

### 4.1. Semen Collection, Proceeding and Exposure to the Tested Substances

Semen samples from 10 dairy bulls (9 Swedish Red and 1 Holstein Friesian), aged 1 to 4 years, were collected twice a week using an artificial vagina according to the standard husbandry method, during October–December 2021, giving a total of 30 ejaculates (3 ejaculates per bull). The bulls were housed under standard husbandry conditions at VikingGenetics, (Skara, Sweden). The semen was extended 1:1 in Andromed extender free of antibiotics (AndroMed^®^ CSS one-step 200mL, Minitüb GmbH, Tiefenbach, Germany), before transportation to the Swedish University of Agricultural Sciences (SLU), overnight at 6 °C, in an insulated box containing a cold pack. Further processing of semen samples and analyses were performed in the laboratories at the Swedish University of Agriculture Sciences, as presented in Figure 3 and described in the text. 

Sperm concentration was measured using a Nucleocounter-SP 100 (Chemometec, Allerød, Denmark) as follows: 50 μL of semen sample were mixed with 5 mL reagent S100 (Chemometec, Allerød, Denmark) to disrupt sperm membranes. This mixture was then loaded into a cassette containing the fluorescent dye propidium iodide. Thereafter, the cassette was inserted into the reading chamber of the fluorescence meter. The sperm concentration appeared on the display. Based on the sperm concentration of each sample, sufficient Andromed extender free of antibiotics was added to give a final concentration of 65 × 10^6^ spermatozoa per mL (spz/mL).

Aliquots of semen samples were exposed to pomegranate, curcumin, or ginger for up to 2 h (Time 1) and 24 h (Time 2) before evaluating sperm quality and an antibacterial effect of the substance. After removing the first aliquots for evaluation at Time 1, the remaining samples were stored at 5 °C until the next day.

Semen collection with an artificial vagina is a routine agricultural practice and, therefore, does not require ethical approval in Sweden; the bulls at the AI (Artificial Insemination) station were not considered to be experimental animals.

### 4.2. Preparation of Pomegranate, Curcumin and Ginger

Ginger extract and curcumin were purchased from Sigma-Aldrich GmbH (Stockholm, Sweden), and commercially available pomegranate powder WellAware Granatäpple; Uppsala, Sweden was used. All tested substances were prepared as described and subsequently added to give a final proportion of 10% and 5% in diluted semen samples containing 30 × 10^6^ spz/mL, to test their antioxidative and antimicrobial properties. Pomegranate powder (20 g) was dissolved in 100 mL distilled water and mixed using a magnetic stirrer for 20 min. The choice of 20% pomegranate solution was based on a pilot experiment in which this proportion was demonstrated to give the best antimicrobial effect on pure bacterial colonies isolated from bull semen [34]. After mixing with a magnetic stirrer, the reconstituted pomegranate solution was centrifuged in 50 mL sterile plastic test tubes for 5 min at 4000× *g* on 22 °C followed by filtration using 0.25 μm pore size filter (Minisart^®^ Syringe Filter, Göttingen, Germany) to remove all potential microorganisms and particles. 

Curcumin (Sigma-Aldrich GmbH, Stockholm, Sweden; 5 mg) was dissolved in 500μL ethanol (70%) in a plastic test tube, and was then diluted to 15 mL with peptone water (Dilucup) to a concentration of 667 μg/mL. Thereafter, 5 mL aliquots were centrifuged for 5 min at 4000× *g* and 22 °C; the supernatant was used.

Ginger extract (250 μL) was dissolved in 6.0 mL dimethyl sulfoxide (DMSO) in a plastic test tube. Thereafter, the mixture was diluted to 13 mL with peptone water and 5 mL aliquots were centrifuged for 5 min at 4000 G and 22 °C. The supernatant was filtered through a 20 Minisart^®^ Syringe Filters (0.25 μm pore size) to eliminate large particles in the ginger dilution. 

In this study, curcumin was dissolved in 70% ethanol according to the suggestion of the manufacturer (Sigma-Aldrich GmbH, Stockholm, Sweden). For ginger, DMSO was suggested as solvent due to its low toxicity, and its ability to dissolve both organic and inorganic compounds [35]. The ginger extract used in this study could not be dissolved in either ethanol or in peptone water, and therefore DMSO was used. The percentages of solvents in the final sperm sample mixtures were 0.33% and 0.17% for ethanol and 4.62% and 2.31% for DMSO, in the 10% and 5% groups, respectively. A previous study [36], evaluated the effect of various solvents on bacterial growth, determining the Minimum Inhibitory Concentration (MIC) of various antimicrobials; the authors stated that ethanol is generally considered safe below 3%. However, the authors did suggest that this cannot be accepted as a general fact for all test organisms. The final concentration of ethanol in semen samples in our study was lower than 0.5%.

### 4.3. Assessment of Sperm Motility Using CASA

Motility evaluation was performed using a SpermVision analyzer (Minitüb GmbH, Tiefenbach, Germany) connected to a Zeiss microscope with a heated stage (38 °C). Semen samples were equilibrated to room temperature before motility analysis. Sperm motility was analyzed in eight fields (at least 850 spermatozoa in total) using the SpermVision software program version 3.8 with settings adjusted for bull spermatozoa, in a 5-μL aliquot of the semen sample on a warm slide covered with an 18 × 18 mm cover slip (VMR, Leuven, Belgium). The following parameters were used to analyze motility: total motility (TM, %), progressive motility (PM, %), curvilinear velocity (VCL, μm/s), average path velocity (VAP, μm/s), straight line velocity (VSL, μm/s), linearity (LIN, as VSL/VCL), straightness (STR, as VSL/VAP), wobble (WOB, as VAP/VCL), beat cross frequency (BCF, Hz), and amplitude of lateral head displacement (ALH, μm). Images were obtained at 200 × magnification using a phase contrast microscope. Particles with an area ranging from 20 to 100 μm^2^ were identified as cells and were included in the analysis. Spermatozoa were considered as immotile if the area under curve (AOC) was < 5, BCF < 0.2, and VSL < 0.2; they were considered to be locally motile if spermatozoa covering a straight-line distance (DSL) was < 4.5.

### 4.4. Flow Cytometry

Flow cytometric (FC) analysis was performed using a FACSVerse^TM^ flow cytometer (BD Biosciences, Becton Dickinson and Company, San Jose, CA, USA). A blue laser emitting at 488 nm and a violet laser emitting at 405 nm were used to excite the fluorescent stains. Green fluorescence (FL1) was detected with a band-pass filter (527/32 nm), as was orange fluorescence (FL2, 586/42 nm); red fluorescence (FL3) was collected using a 700/54 nm band-pass filter, while blue fluorescence (FL5) was collected using a 528/45 nm band-pass filter. The data obtained in the sperm chromatin structure assay were further analyzed using FCS Express 5 software (De Novo, Glendale, CA, USA).

#### 4.4.1. Assessment of Membrane Integrity Using Flow Cytometry 

Evaluation of sperm plasma membrane integrity was performed using SYBR14 and propidium iodide (PI) (Live-Dead Sperm Viability Kit L-7011; Invitrogen, Eugene, OR, USA). Aliquots of each sample were adjusted to a sperm concentration of approximately 2 × 10^6^ sperm cells/mL in 300 μL of buffer B (patent pending; J.M. Morrell and H. Rodriguez-Martinez). Thereafter, 0.5 μL of 1 mM SYBR14 was diluted 50 times with Buffer B and 1.2 μL was added to each sperm sample. The sperm samples were also stained with 3 μL of 2.4 mM PI. The stained aliquots were incubated at 38 °C for 10 min before evaluation was performed and the proportions of membrane intact, membrane damaged, and intermediate populations were enumerated.

#### 4.4.2. Assessment of Reactive Oxygen Species

Hydroethidine (HE; Invitrogen, Thermo Fisher Scientific, Eugene, OR, USA) and 20, 70 -dichlorodihydrofluorescein diacetate (DCFDA; Invitrogen, Thermo Fisher Scientific, Eugene, OR, USA) were used to detect superoxide (O^2−^) and hydrogen peroxide (H_2_O_2_), respectively, while Hoechst 33258 (HO) was added to permit the simultaneous differentiation of living and dead cells. Aliquots (300 μL) of semen extended to a concentration of approximately 2 × 10^6^ spermatozoa/mL (spz/mL) with Buffer B were stained using each of the following: 3 µL of HO (40 mM), 3 µL HE (40 mM), and 3 µL DCFDA, (2 mM). The samples were gently mixed and incubated at 38 °C for 30 min before analysis. Using dot-plots for HO/HE and HO/DCFDA, the following populations were quantified: ROS Live SO^−^; ROS Live SO^+^; ROS Dead SO^+^; ROS Dead H_2_O_2_^−^; ROS Dead H_2_O_2_^+^; ROS Live H_2_O_2_^−^; and ROS Live H_2_O_2_^+^.

#### 4.4.3. Mitochondrial Membrane Potential 

Sperm samples were diluted with Buffer B in order to obtain a concentration of 2.5 × 10^6^ spz/mL. In order to evaluate the mitochondrial potential, 1.2 µL JC-1 (stock 3 mM) was mixed with 300 µL sperm aliquot and incubated for at least 30 min at 38 °C before analyses. JC-1 fluorescence was measured in the FL1 and FL2 channels of the flow cytometer. A total number of 10,000 cells was evaluated and classified as percentages in two distinct groups: sperm cells with high respiratory activity emitting orange florescence, and low respiratory activity emitting green florescence. 

#### 4.4.4. Sperm Chromatin Structure Assay

Chromatin integrity was evaluated using the sperm chromatin structure assay (SCSA). The test uses the metachromatic dye acridine orange (AO) to assess the susceptibility of sperm deoxyribonucleic acid (DNA) to acid-induced denaturation. The DNA fragmentation index (%DFI) is calculated and expressed as the proportion of cells with a high ratio of denatured, single stranded DNA; %DFI = (red fluorescence/[green fluorescence + red fluorescence]) × 100. The procedure, media preparation, buffers, and solutions used in the assay have been described in detail previously [37]. Briefly, 50 μL of the semen sample was mixed with the same amount of Tris, sodium chloride, and ethylene-diaminetetraacetic acid buffer (TNE buffer) and immediately transferred to a liquid nitrogen container for snap-freezing. The samples were stored at −80 °C until analysis. They were thawed on ice, an aliquot (10 µL) was mixed with 90 µL of TNE, and 200 μL of acid-detergent solution. Exactly 30 s later, the sample was stained by adding 600 µL of AO staining solution. The stained samples were analyzed within 3–5 min of AO staining.

### 4.5. Bacterial Quantification 

The total number of viable aerobic bacteria in the semen samples was analyzed according to NMKL 86, 5 Ed., 2013, with slight modifications. In brief, 1 mL of each sperm sample was transferred to 1 mL of diluent. The samples were homogenized using a vortex (Saveen & Werner, Malmö, Sweden). The diluent was prepared at SLU using 1 g peptone and 8.5 g NaCl per liter Milli-Q H2O and was autoclaved at 121 °C for 15 min. Plate count agar (PCA) (Oxoid, Basingstoke, UK) was melted in boiled water and thereafter placed in a 48 °C water bath to keep it liquid until required. A 2-fold serial dilution was prepared until 1/4 of the sperm sample was obtained, 1.0 mL from each dilution was pour plated into a petri dish measuring 9 cm in diameter and mixed with 10–15 mL melted PCA. When the agar in the petri dishes had solidified, a further 10–15 mL of PCA was added to each dish as an over layer to avoid swarming and facilitate enumeration, and the petri dishes were moved gently to distribute the bacteria evenly over the plates. After solidification of the agar, the plates were incubated at 30 °C for 72 h. The number of viable bacteria was quantified from the plates with more than 10 but less than 250 colonies. The total number of CFU was calculated from three successive dilutions using a colony counter (Gerber instruments, Im Langhag, Switzerland).

The total bacterial count was assessed in aliquots taken 2 h after adding the plant-based substances (Time 1) and again after 24 h (Time 2).

### 4.6. Statistical Analysis

The model contained bulls, exposure time, and substances as fixed factors. Interactions were tested between different bulls, substances and time, but were removed if there was no significance. Differences between control and tested substances for CASA and Flow cytometry sperm quality parameters were analysed by MANOVA Dunnett’s multiple comparison test (IBM Statistic SPSS 26), with three levels of significance (*p* < 0.05, *p* < 0.01 and *p* < 0.001). Data are presented as mean ± standard deviation (SD). Descriptive statistics were used to present bacteriological results.

## 5. Conclusions

A reduction in bacterial count was present for all tested substances. The tested substances in different concentrations did not themselves have a negative effect on sperm quality parameters, considering the deleterious effect of DMSO as a solvent. Curcumin at a concentration of 5% was the only substance that had a slight positive effect on sperm motility parameters and a noticeable bacterial reduction (32%), at Time 1. Further studies on combinations of plant-based substances could be conducted to decrease individual toxic effects and increase beneficial effects on sperm quality and an antibacterial effect on bull semen microbiome.

## Figures and Tables

**Figure 1 ijms-24-03435-f001:**
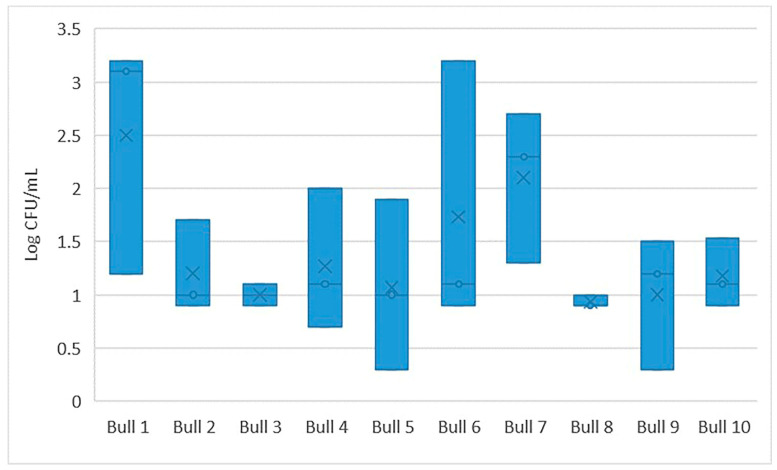
Number of bacteria (Log CFU/mL) in control samples (*n* = 3) for individual bulls at Time 1.

**Figure 2 ijms-24-03435-f002:**
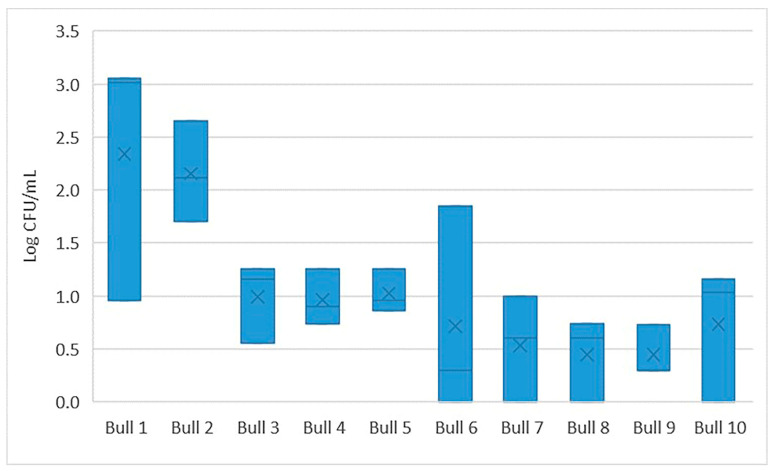
Number of bacteria (Log CFU/mL) in control samples (*n* = 3) for individual bulls at Time 2.

**Figure 3 ijms-24-03435-f003:**
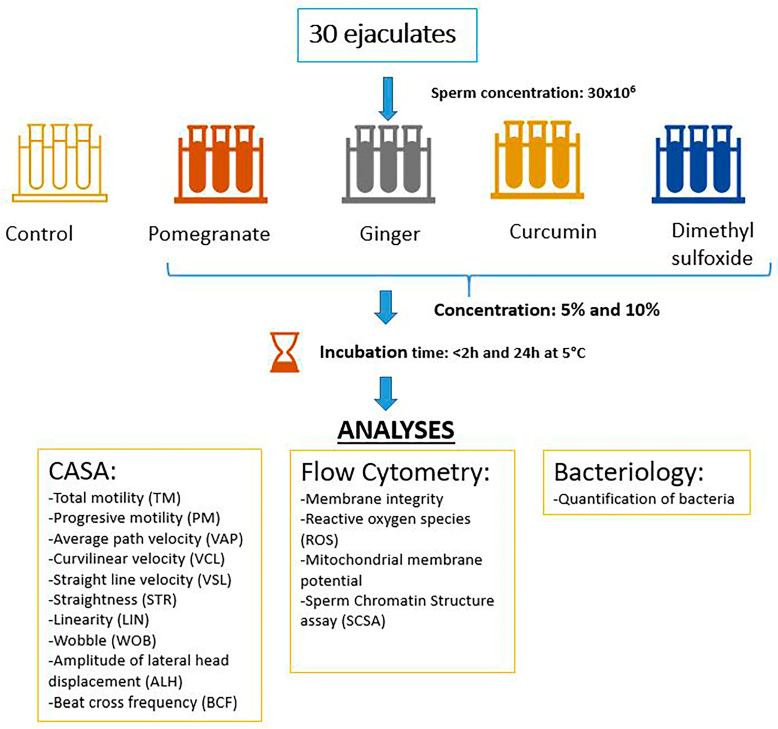
Incubation time and analyses after adding various plant-based substances to bull semen 24 h after collection. In total, semen samples from 10 bulls were used (three ejaculates per bull).

**Table 1 ijms-24-03435-t001:** CASA parameters of bull semen samples after exposure to various plant-based substances at Time 1.

	Control	P 10%	P 5%	G 10%	G 5%	C 10%	C 5%	D 10%	D 5%
TM	74.95 ± 13.34	74.91 ± 12.83	77.50 ± 11.01	64.49 ± 15.42 ^b^	68.61 ± 11.90	73.43 ± 13.93	72.54 ± 12.15	61.96 ± 14.68 ^c^	63.16 ± 15.92 ^b^
PM	72.56 ± 13.62	72.04 ± 13.58	74.90 ± 11.44	61.61 ± 15.48 ^b^	65.66 ± 12.23	70.17 ± 14.29	70.04 ± 12.28	58.90 ± 14.87 ^c^	60.83 ± 15.78 ^b^
VAP	64.44 ± 10.13	60.45 ± 6.65	65.19 ± 6.91	62.63 ± 8.58	58.81 ± 7.72 ^a^	64.28 ± 10.27	56.48 ± 9.28 ^b^	56.13 ± 6.57 ^b^	50.11 ± 5.17 ^c^
VCL	119.21 ± 24.20	115.48 ± 14.44	121.07 ± 17.79	122.03 ± 20.76	107.58 ± 17.76	124.08 ± 25.62	101.93 ± 21.30 ^b^	105.05 ± 15.56 ^a^	89.37 ± 11.29 ^c^
VSL	50.23 ± 8.35	45.28 ± 6.80 ^a^	51.28 ± 5.96	43.05 ± 4.79 ^c^	44.95 ± 6.09 ^b^	46.35 ± 7.26	43.12 ± 7.95 ^c^	39.71 ± 4.85 ^c^	37.24 ± 4.56 ^c^
STR	0.78 ± 0.05	0.74 ± 0.06 ^b^	0.78 ± 0.046	0.69 ± 0.05 ^c^	0.76 ± 0.04	0.72 ± 0.06 ^c^	0.76 ± 0.04	0.70 ± 0.05 ^c^	0.74 ± 0.05 ^b^
LIN	0.42 ± 0.06	0.39 ± 0.06 ^b^	0.42 ± 0.06	0.35 ± 0.04 ^c^	0.42 ± 0.04	0.38 ± 0.06 ^c^	0.42 ± 0.05	0.37 ± 0.04 ^c^	0.41 ± 0.04
WOB	0.54 ± 0.04	0.52 ± 0.04 ^a^	0.54 ± 0.05	0.51 ± 0.03 ^c^	0.54 ± 0.03	0.52 ± 0.04 ^a^	0.55 ± 0.04	0.53 ± 0.03	0.56 ± 0.03
ALH	4.38 ± 0.95	4.40 ± 0.62	4.35 ± 0.70	5.09 ± 0.94 ^c^	4.18 ± 0.76	4.84 ± 1.02 ^a^	4.01 ± 0.71	4.55 ± 0.78	3.68 ± 0.65 ^c^
BCF	25.24 ± 2.79	7.28 ± 10.03 ^b^	23.21 ± 20.44 ^a^	18.96 ± 14.33 ^c^	29.95 ± 18.6 ^b^	32.52 ± 25.01 ^c^	40.04 ± 21.86 ^b^	35.47 ± 19.09 ^c^	35.37 ± 16.97 ^c^

Abbreviations: P—pomegranate powder, G—ginger, C—curcumin, D—Dimethyl sulfoxide, TM—Total motility, PM—Progressive motility, VCL—curvilinear velocity, VAP—average path velocity, VSL—straight line velocity, LIN—linearity, STR—straightness, WOB—wobble, BCF—beat cross frequency, ALH—amplitude of lateral head displacement; Superscript letters denote significant differences within a row compared with control: a—*p* < 0.05; b—*p* < 0.01; c—*p* < 0.001.

**Table 2 ijms-24-03435-t002:** CASA parameters of bull semen samples after exposure to various plant-based substances at Time 2.

	Control	P 10%	P 5%	G 10%	G 5%	C 10%	C 5%	D 10%	D 5%
TM	37.75 ± 22.95	7.28 ± 10.04 ^c^	23.21 ± 20.44 ^b^	18.96 ± 14.33 ^c^	29.95 ± 8.60	32.52 ± 25.01	40.06 ± 21.86	35.47 ± 19.09	35.37 ± 16.97
PM	33.13 ± 23.66	5.19 ± 8.78 ^c^	19.01 ± 20.12 ^b^	15.64 ± 13.21 ^c^	26.29 ± 17.63	29.81 ± 24.06	37.09 ± 21.28	32.31 ± 18.75	32.04 ± 16.66
VAP	41.43 ± 11.45	24.48 ± 14.20 ^c^	32.33 ± 11.77 ^b^	35.43 ± 10.34	37.41 ± 9.41	41.66 ± 6.58	40.30 ± 7.36	40.50 ± 8.01	36.94 ± 9.85
VCL	74.87 ± 24.16	38.55 ± 22.08 ^c^	55.79 ± 20.13 ^c^	60.17 ± 19.36 ^b^	65.44 ± 18.11	72.11 ± 15.54	69.23 ± 13.88	71.83 ± 17.46	65.61 ± 19.22
VSL	30.20 ± 9.41	18.18 ± 11.51 ^c^	23.74 ± 9.99 ^a^	24.06 ± 7.79 ^b^	27.40 ± 7.40	30.38 ± 5.28	30.51 ± 6.57	27.53 ± 6.47	26.09 ± 7.32
STR	0.71 ± 0.07	0.61 ± 0.29 ^a^	0.70 ± 0.15	0.65 ± 0.14	0.73 ± 0.07	0.72 ± 0.05	0.75 ± 0.05	0.67 ± 0.07	0.68 ± 0.15
LIN	0.40 ± 0.06	0.40 ± 0.22	0.41 ± 0.12	0.39 ± 0.10	0.42 ± 0.06	0.43 ± 0.06	0.44 ± 0.05	0.38 ± 0.06	0.38 ± 0.10
WOB	0.56 ± 0.05	0.53 ± 0.26	0.56 ± 0.13	0.57 ± 0.13	0.57 ± 0.05	0.58 ± 0.08	0.58 ± 0.04	0.57 ± 0.05	0.54 ± 0.11
ALH	3.16 ± 0.95	2.09 ± 1.40 ^c^	2.61 ± 1.18	2.82 ± 0.91	2.78 ± 0.83	3.12 ± 0.62	2.85 ± 0.61	3.26 ± 0.64	2.81 ± 0.89
BCF	20.21 ± 2.90	14.14 ± 7.78 ^c^	17.52 ± 4.25	18.31 ± 4.86	19.65 ± 2.98	20.52 ± 2.92	20.81 ± 2.39	19.06 ± 2.37	18.89 ± 4.49

Abbreviations: P—pomegranate powder, G—ginger, C—curcumin, D—Dimethyl sulfoxide, TM—Total motility, PM—Progressive motility, VCL—curvilinear velocity, VAP—average path velocity, VSL—straight line velocity, LIN—linearity, STR—straightness, WOB—wobble, BCF—beat cross frequency, ALH—amplitude of lateral head displacement; Superscript letters within a row denote significant differences compared with control: a—*p* < 0.05; b—*p* < 0.01; c—*p* < 0.001.

**Table 3 ijms-24-03435-t003:** Flow cytometry results of bull semen samples at Time 1 after exposure to various plant-based substances.

	Control	P 10%	P 5%	G 10%	G 5%	C 10%	C 5%	D 10%	D 5%
MI Living	78.77 ± 12.65	75.97 ± 13.17	76.16 ± 13.30	11.03 ± 11.86 ^c^	26.92 ± 17.29 ^c^	75.24 ± 13.90	75.97 ± 14.31	8.56 ± 9.27 ^c^	24.06 ± 14.34 ^c^
MI Dying	3.75 ± 6.96	1.76 ± 3.71	0.82 ± 0.54	23.76 ± 13.95 ^c^	15.07 ± 12.48 ^c^	0.91 ± 0.84	0.59 ± 0.30	18.24 ± 13.65 ^c^	10.07 ± 7.77 ^b^
MI Dead	17.48 ± 11.50	22.27 ± 12.53	23.02 ± 13.24	65.21 ± 15.76 ^c^	58.01 ± 17.78 ^c^	23.85 ± 13.67	23.44 ± 14.28	73.20 ± 14.76 ^c^	65.87 ± 15.16 ^c^
ROS Live SO^−^	60.83 ± 16.04	59.50 ± 12.32	63.26 ± 13.19	10.15 ± 13.40 ^c^	19.15 ± 14.43 ^c^	59.79 ± 15.24	61.10 ± 14.15	7.87 ± 10.03 ^c^	18.55 ± 15.39 ^c^
ROS Live SO^+^	12.27 ± 11.50	10.35 ± 4.53	9.34 ± 3.60	58.62 ± 18.01 ^c^	50.06 ± 17.04 ^c^	10.91 ± 5.15	9.93 ± 4.26	59.32 ± 17.46 ^c^	48.74 ± 20.20 ^c^
ROS Dead SO^+^	21.22 ± 12.62	24.24 ± 12.12	22.41 ± 12.10	22.14 ± 12.35	21.35 ± 11.85	23.73 ± 13.50	23.45 ± 12.66	23.40 ± 12.21	23.07 ± 11.94
ROS Dead H_2_O_2_^−^	28.08 ± 12.82	31.35 ± 12.29	29.32 ± 12.44	48.33 ± 13.30 ^c^	47.17 ± 12.53 ^c^	31.52 ± 14.10	30.87 ± 13.13	50.14 ± 14.39 ^c^	46.60 ± 11.22 ^c^
ROS Dead H_2_O_2_^+^	0.11 ± 0.26	0.03 ± 0.02	0.03 ± 0.01	0.06 ± 0.03 ^c^	0.04 ± 0.02 ^c^	0.10 ± 0.11	0.05 ± 0.04	0.03 ± 0.01 ^c^	0.43 ± 2.23 ^c^
ROS Live H_2_O_2_^−^	71.8 ± 12.89	68.61 ± 12.30	70.65 ± 12.44	51.61 ± 13.31 ^c^	52.78 ± 12.53 ^c^	68.37 ± 14.08	69.07 ± 13.12	49.82 ± 14.39 ^c^	52.77 ± 11.28 ^c^
ROS Live H_2_O_2_^+^	0.01 ± 0.01	0.01 ± 0.01	0.01 ± 0.02	0.00 ± 0.01	0.00 ± 0.01	0.01 ± 0.01	0.01 ± 0.02	0.00 ± 0.01	0.19 ± 1.01
MMP High RA	67.23 ± 13.07	59.12 ± 19.97 ^a^	65.99 ± 15.57	12.24 ± 11.07 ^c^	22.28 ± 14.52 ^c^	66.88 ± 13.72	67.62 ± 13.91	11.63 ± 13.36 ^c^	23.33 ± 13.57 ^c^
MMP Low RA	32.77 ± 13.07	40.88 ± 19.97 ^a^	34.01 ± 15.57	87.76 ± 11.07 ^c^	77.72 ± 14.52 ^c^	33.12 ± 13.72	32.38 ± 13.91	88.37 ± 13.36 ^c^	76.67 ± 13.57 ^c^
DFI %	9.27 ± 4.68	22.20 ± 11.23 ^c^	22.11 ± 11.77 ^c^	9.34 ± 4.04	9.62 ± 4.22	9.50 ± 4.38	9.41 ± 4.32	9.75 ± 4.43	9.49 ± 4.25

Abbreviations: Control; P—pomegranate powder, G—ginger, C—curcumin, D—Dimethyl sulfoxide; MI—Membrane integrity, ROS—Reactive oxygen species, SO—Superoxide, MMP—Mitochondrial membrane potential, RA—Respiratory activity, DFI—DNA fragmentation index; Superscript letters within a row denote significant differences compared with control: a—*p* < 0.05; b—*p* < 0.01; c—*p* < 0.001.

**Table 4 ijms-24-03435-t004:** Flow cytometry results of bull semen samples at Time 2 after exposure to various plant-based substances.

	Control	P 10%	P 5%	G 10%	G 5%	C 10%	C 5%	D 10%	D 5%
MI Living	76.76 ± 13.72	68.57 ± 17.20 ^a^	73.03 ± 14.14	9.28 ± 11.73 ^c^	27.60 ± 18.03 ^c^	73.31 ± 14.28	74.05 ± 13.53	8.07 ± 9.26 ^c^	25.54 ± 17.33 ^c^
MI Dying	1.98 ± 2.42	3.62 ± 9.15	1.97 ± 3.50	14.68 ± 10.55 ^c^	12.58 ± 7.93 ^c^	1.43 ± 1.05	0.97 ± 1.10	8.79 ± 5.70 ^b^	8.80 ± 14.72 ^b^
MI Dead	21.26 ± 14.38	27.81 ± 14.45	25.00 ± 13.97	76.04 ± 15.42 ^c^	59.82 ± 19.70 ^c^	25.26 ± 14.00	24.98 ± 13.42	83.14 ± 11.15 ^c^	65.66 ± 20.00 ^c^
ROS Live SO^−^	59.89 ± 14.35	52.06 ± 15.25	52.77 ± 16.65	8.61 ± 12.06 ^c^	21.55 ± 16.54 ^c^	57.28 ± 17.55	56.83 ± 18.07	7.66 ± 10.65 ^c^	17.85 ± 15.60 ^c^
ROS Live SO^+^	12.00 ± 6.39	15.40 ± 6.47	15.00 ± 7.68	51.99 ± 16.36 ^c^	44.12 ± 17.35 ^c^	11.90 ± 5.5	11.74 ± 5.23	54.67 ± 16.35 ^c^	45.98 ± 16.37 ^c^
ROS Dead SO^+^	21.63 ± 12.9	24.30 ± 13.33	25.81 ± 12.46	27.28 ± 11.41	23.39 ± 11.53	25.78 ± 15.37	25.14 ± 15.13	26.63 ± 11.26	24.50 ± 12.08
ROS Dead H_2_O_2_^−^	31.01 ± 14.51	37.77 ± 4.57	38.03 ± 14.5	62.71 ± 12.97 ^c^	53.33 ± 12.04 ^c^	41.64 ± 19.64	36.59 ± 17.94	59.21 ± 14.34 ^c^	56.12 ± 12.66 ^c^
ROS Dead H_2_O_2_^+^	0.09 ± 0.14	0.03 ± 0.02 ^b^	0.03 ± 0.02 ^b^	0.05 ± 0.03	0.04 ± 0.01 ^a^	0.11 ± 0.14	0.06 ± 0.07	0.02 ± 0.01 ^c^	0.02 ± 0.01 ^c^
ROS Live H_2_O_2_^−^	68.90 ± 14.54	62.19 ± 14.58	61.93 ± 14.5	37.23 ± 12.97 ^c^	46.30 ± 11.97 ^c^	58.25 ± 19.69 ^b^	63.34 ± 17.94	40.77 ± 14.34 ^c^	43.85 ± 12.66 ^c^
ROS Live H_2_O_2_^+^	0.00 ± 0.01	0.00 ± 0.01	0.01 ± 0.01	0.00 ± 0	0.00 ± 0	0.00 ± 0	0.01 ± 0.01	0.00 ± 0.01	0.00 ± 0.01
MMP High RA	40.57 ± 26.34	26.59 ± 18.22 ^b^	34.80 ± 21.45	5.82 ± 5.03 ^c^	14.82 ± 15.06 ^c^	36.93 ± 28.19	42.03 ± 25.88	9.80 ± 16.63 ^c^	21.05 ± 23.28 ^c^
MMP Low RA	59.43 ± 26.34	73.41 ± 18.22 ^b^	65.20 ± 21.45	94.18 ± 5.03 ^c^	85.18 ± 15.06 ^c^	63.07 ± 28.19	57.97 ± 25.88	90.20 ± 16.63 ^c^	78.95 ± 23.28 ^c^
DFI %	20.46 ± 26.79	23.56 ± 12.56	22.53 ± 11.37	36.99 ± 32.54 ^a^	35.87 ± 33.61	25.21 ± 30.41	20.17 ± 26.51	39.70 ± 29.64 ^b^	41.10 ± 32.70 ^b^

Abbreviations: Control; P—pomegranate powder, G—ginger, C—curcumin, D—Dimethyl sulfoxide; MI—Membrane integrity, ROS—Reactive oxygen species, SO—Superoxide, MMP—Mitochondrial membrane potential, RA—Respiratory activity, DFI—DNA fragmentation index; Superscript letters denote significant differences within a row compared with control: a—*p* < 0.05; b—*p* < 0.01; c—*p* < 0.001.

**Table 5 ijms-24-03435-t005:** Number of bacteria (CFU/mL) in the semen before (control) and after exposure of different substances at Time 1 and Time 2 at 5 °C.

Substances	Time 1 Mean (Range) Log CFU/mL	Time 1 Reduction ^a^ CFU/mL (%)	Time 2 Mean (Range) Log CFU/mL	Time 2 Reduction ^a^ CFU/mL (%)	Reduction Time 1 vs. Time 2 CFU/mL (%)
Control	2.27 (0.3–3.2)		2.00 (0.0–3.1)		46%
P 5%	2.16 (0.0–3.3)	22%	2.01 (0.0–3.3)	−3%	28%
P 10%	2.15 (0.0–3.1)	25%	1.98 (0.0–3.2)	5%	31%
G 5%	2.12 (0.0–3.2)	29%	1.91 (0.0–3.0)	20%	39%
G 10%	2.03 (0.0–3.2)	42%	1.84 (0.0–3.0)	31%	35%
C 5%	2.10 (0.0–3.2)	32%	1.91 (0.0–3.1)	20%	37%
C 10%	2.18 (0.0–3.3)	20%	1.98 (0.0–3.2)	6%	37%
D 5%	1.93 (0.0–3.1)	54%	1.79 (0.0–2.9)	38%	27%
D 10%	2.13 (0.0–3.1)	28%	1.74 (0.0–2.9)	46%	59%

Abbreviations: P—pomegranate powder, G—ginger, C—curcumin, D—dimethyl sulfoxide, CFU—Colony forming units; ^a^ Note: The percentage reduction between control and individual tested substances is calculated on the absolute values and not on the log-transformed values and presented in columns Time 1 Reduction and Time 2 Reduction. Bacterial reduction during storage is calculated on the absolute values within the row; presented in column Reduction Time 1 vs. Time 2.

## Data Availability

The data presented in this study are available on request from the corresponding author.

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
