# Peer review of "Effect of Some Plant-Based Substances on Microbial Content and Sperm Quality Parameters of Bull Semen"

_ijms, 2023, doi:10.3390/ijms24043435_

Round 1

Reviewer 1 Report

1- ms format should be arranged to journal rules.,

2- please write abstract again according to your results.

3- ıntroduction can be extended with some literatures of curcumin:

Bucak MN, Sarıözkan S, Tuncer PB, Sakin F, Ateşşahin A, Kulaksız R, Çevik M (2010). The effect of antioxidants on post-thawed Angora goat (Capra hircus ancryrensis) sperm parameters, lipid peroxidation and antioxidant activities. Small Rumin Res, 89, 24-30

Bucak MN, Başpınar N, Tuncer PB, Sarıözkan S, Akalın PP, Çoyan K (2012). Effects of curcumin and dithioerythritol on frozen-thawed bovine semen. Andrologia, 44, 102–109.

4- in material method: 

please extend sperm extending part.

please explain detaily for reader: part of Preparation of pomegranate, curcumin and ginger

5- please re-write the results in good way detaily with p values.

6-please correct the table format

7- please add dna test figure

Author Response

  • Comment: 1 ms format should be arranged to journal rules.
  • Authors: The changes have been made
  • Comment: 2- please write abstract again according to your results.
  • Authors: We have actually followed Instructions for authors that explain in Abstract for “Results: Summarize the article's main findings”, which we have done. We state that Curcumin 5% was the only substance that had a slight positive effect on sperm kinematics, but we have now added that the other substances had a deleterious effect on sperm quality. Bacterial reduction was also presented.
  • Comment: 3- ıntroduction can be extended with some literatures of curcumin:

Bucak MN, Sarıözkan S, Tuncer PB, Sakin F, Ateşşahin A, Kulaksız R, Çevik M (2010). The effect of antioxidants on post-thawed Angora goat (Capra hircus ancryrensis) sperm parameters, lipid peroxidation and antioxidant activities. Small Rumin Res, 89, 24-30

Bucak MN, Başpınar N, Tuncer PB, Sarıözkan S, Akalın PP, Çoyan K (2012). Effects of curcumin and dithioerythritol on frozen-thawed bovine semen. Andrologia, 44, 102–109.

  • Authors: It is added, please see line 66-69
  • Comment: 4- in material method: 

please extend sperm extending part.

  • Authors: Please see line 275-277

please explain detaily for reader: part of Preparation of pomegranate, curcumin and ginger.

  • Authors: We have now provided a more detailed explanation of the reasons for using different solvents for different substances. We have added the proportion of the solvents in the mixture with substances and in in semen samples. The final proportions of substances and solvent in semen samples was also added. Please see Lines 308-319.
  • Comment: 5- please re-write the results in good way detaily with p values.
  • Authors: The different p values are already presented in Tables 1-4 as Superscript letters (a, b and c) and explained in the footnotes. Because there are 10 CASA and 13 Flow Cytometry parameters and 9 groups, we found Table form more suitable to present this huge amount of data, as mentioned by another Reviewer. For example, in Table 3, substances G 10%, G 5%, D 10% and D 5% had a negative influence on all FC parameters except chromatin integrity at 0h; we did not mention all of these p values again in the text since they are all presented in the Table. Similarly, the p values are presented in Table 4 but not repeated in the text.
  • Comment: 6-please correct the table format
  • Authors: The format is corrected.
  • Comment: 7- please add dna test figure
  • The Sperm chromatin structure assay was presented as DNA fragmentation index (DFI %) in Table 3. and Table 4. Can you please specify what kind of figure would you like to be presented?

Reviewer 2 Report

Review to manuscript IJMS-2156236 Effect of some plant-based substances on microbial content and

sperm quality parameters of bull semen

In the last decades several ingredients of the semen extenders were found to be as potential danger in case of the future usage. It can be seen that animal origin components  like proteins could be replaced by plant based ones. The next focus of researches is on antibiotics which are used long time ago to keep safe the doses from harmful microbes. However the threat of increasing antimicrobial resistance makes press on in replacement of them by alternative agents or methods. One of the options is using plant based extracts. Most of these extracts have several advantages in biological systems and formerly widely used as antioxidants during cryopreservation of semen samples. However lots of them have additional antimicrobial effects too thus they are good candidates for replacements of antibiotics. The optimal dosages and solvent/vehicle of them should be determined.

Scope of the manuscript is to test pomegranate, ginger and curcumin as antimicrobial agents on bull semen preservation.

Title of the manuscript is clear and scope of it is fulfill the journal’s interest. Scientific background of the work is well described and gives a strong basis of the theme’s impact.

Materials and methods are fully detailed all the treatment protocols and used methods. May be some more information about the bulls are necessary (breeds, purpose etc.). All the semen concentrations should be corrected in case of x 106 , where it is written x106. In subchapter 4.4.2. line 330 it was written thawed samples, may be false? In CASA investigation it was not used special chamber to load and evaluate the samples?

Results were demonstrated in 5 tables and 2 figures which are necessary for better understanding of the collected huge amont of data.

The main finding is that the Curcumin 5% substrate has a very strong antimicrobial effect in samples at time 1 without any detrimental influence on semen kinematics and cell structural integrities. However further trials are needed with different active substrates’ solvents and combined usage of these plant base dantimicrobial agents.

References should be unified in case of Journals’ titles.

After these minor changes It is recommended to publish the manuscript.

Author Response

In the last decades several ingredients of the semen extenders were found to be as potential danger in case of the future usage. It can be seen that animal origin components like proteins could be replaced by plant based ones. The next focus of researches is on antibiotics which are used long time ago to keep safe the doses from harmful microbes. However the threat of increasing antimicrobial resistance makes press on in replacement of them by alternative agents or methods. One of the options is using plant based extracts. Most of these extracts have several advantages in biological systems and formerly widely used as antioxidants during cryopreservation of semen samples. However lots of them have additional antimicrobial effects too thus they are good candidates for replacements of antibiotics. The optimal dosages and solvent/vehicle of them should be determined.

Scope of the manuscript is to test pomegranate, ginger and curcumin as antimicrobial agents on bull semen preservation.

Title of the manuscript is clear and scope of it is fulfill the journal’s interest. Scientific background of the work is well described and gives a strong basis of the theme’s impact.

  • Comment: Materials and methods are fully detailed all the treatment protocols and used methods. May be some more information about the bulls are necessary (breeds, purpose etc.). All the semen concentrations should be corrected in case of x 106, where it is written x106. In subchapter 4.4.2. line 330 it was written thawed samples, may be false? In CASA investigation it was not used special chamber to load and evaluate the samples?
  • Authors: Thank you for the comments. Semen concentration are corrected through the manuscript and “thawed” has been deleted. We added description about bull breeds and purpose (please see Line 256-257). As far as CASA analyses are concerned, we did not use a special chamber. We use a glass slide with a specific volume (5 µL semen) under an 18x18 mm coverslip for all assessments, as already specified in lines 325-327.
  • Comment: Results were demonstrated in 5 tables and 2 figures which are necessary for better understanding of the collected huge amont of data.
  • Authors: Thank you

The main finding is that the Curcumin 5% substrate has a very strong antimicrobial effect in samples at time 1 without any detrimental influence on semen kinematics and cell structural integrities. However further trials are needed with different active substrates’ solvents and combined usage of these plant base dantimicrobial agents.

  • Comment: References should be unified in case of Journals’ titles.
  • Authors: The changes have been made.

After these minor changes It is recommended to publish the manuscript

Reviewer 3 Report

Line36: I believe that you meant „consumption“

Line 47: maybe it would be better to specify that it is European legislation (European Council Directive)

Line 60: correct – Curcuma longa and change to italics.

Line 63: Staphylococcus aureus – italics

…Change in whole manuscript Latin names of species to italics.

Line 67: Schulze et al.

Figure 3: Correct the bottom margin – make it visible.

Line 272: correct superscript and change “x” - 30 × 106 …check in whole manuscript

Line 278: 4000 × G (check whole methodology)

Line 359: Change the section's name to reflect methodology..for instance bacterial quantification or total bacterial count..

Line 87: correct distance from margin..same for line 96..each table title

Line 129: was observed

Lines 167-171: I don't think this is the right argument..I suppose that each group was measured after 48 hours from the collection.

Check instruction for authors.. correct whole manuscript..also references

Can you please explain why did you dissolve each substance in different solvents? What was the percentage of each used solvent in final mixture? Why does your study design include DMSO group but not ethanol group?

Did you analyze microbial quality?

Try to unify all tables. If there will be 2 lines in 1 table cell, make it in whole table. However, it would be ideal if it fits in 1 line.

Author Response

  • Comment: Line36: I believe that you meant „consumption“
  • Authors: Yes, and it is corrected (Line 40).
  • Comment: Line 47: maybe it would be better to specify that it is European legislation (European Council Directive)
  • Authors: It is added now, see Line 51.
  • Comment: Line 60: correct – Curcuma longa and change to italics.
  • Authors: It is corrected (Line 65)
  • Comment: Line 63: Staphylococcus aureus – italics

…Change in whole manuscript Latin names of species to italics.

  • Authors: The changes have been made.
  • Comment: Line 67: Schulze et al.
  • Authors: It is corrected (Line 73)
  • Comment: Figure 3: Correct the bottom margin – make it visible.
  • Authors: The Figure 3 is replaced with new one, whit the visible bottom margin.
  • Comment: Line 272: correct superscript and change “x” - 30 × 106 …check in whole manuscript
  • Authors: It is corrected, please see Line 290.
  • Comment: Line 278: 4000 × G (check whole methodology).
  • Authors: It is corrected, see Line 296.
  • Comment: Line 359: Change the section's name to reflect methodology..for instance bacterial quantification or total bacterial count..
  • Authors: The title 4.5. has been changed. Please see Line 389.
  • Comment: Line 87: correct distance from margin..same for line 96..each table title. Line 129: was observed
  • Authors: It is corrected.
  • Comment: Lines 167-171: I don't think this is the right argument..I suppose that each group was measured after 48 hours from the collection.
  • Authors: Thank you for the comment. We have removed these sentences.
  • Comment: Check instruction for authors.. correct whole manuscript..also references
  • Authors: Journal template have been used for manuscript submission and authors have tried to follow instructions and apologize if any mistakes still appears. All changes have been made as suggested.
  • Comment: Can you please explain why did you dissolve each substance in different solvents? What was the percentage of each used solvent in final mixture? Why does your study design include DMSO group but not ethanol group?
  • Authors: This new paragraph answering on all questions has been added in to manuscript. Please see Line 308-319.
  • Comment: Did you analyze microbial quality?
  • Authors: No, we did not analyzed microbial quality. That was not the aim of this study; the focus was to analyse antibacterial effect by evaluating bacterial reduction and to determine sperm quality after adding known concentrations of plant-based substances..
  • Comment: Try to unify all tables. If there will be 2 lines in 1 table cell, make it in whole table. However, it would be ideal if it fits in 1 line.
  • The changes have been made.

Round 2

Reviewer 3 Report

There were made great changes which definitely helped the manuscript clarity. Thank you. I just recommend to improve quality of figures. Please, upload each figure with at least 300dpi. 

Author Response

We have increased the resolution of the Figures. Note the old Figures were deleted before inserting the new ones